# Infection with Jujube Witches’ Broom Phytoplasma Alters the Expression Pattern of the *Argonaute* Gene Family in *Ziziphus jujuba*

**DOI:** 10.3390/microorganisms13030658

**Published:** 2025-03-14

**Authors:** Jia Yao, Zesen Qiao, Ziming Jiang, Xueru Zhao, Ziyang You, Wenzhe Zhang, Jiancan Feng, Chenrui Gong, Jidong Li

**Affiliations:** 1College of Forestry, Henan Agricultural University, Zhengzhou 450046, China; yaojia@stu.henau.edu.cn (J.Y.); zesenqiao@163.com (Z.Q.); jiangziming@stu.henau.edu.cn (Z.J.); xrzhao@stu.henau.edu.cn (X.Z.); dzzzacrkwd@outlook.com (Z.Y.); wenzhezhang@stu.henau.edu.cn (W.Z.); 2College of Horticulture, Henan Agricultural University, Zhengzhou 450046, China; jcfeng@henau.edu.cn

**Keywords:** jujube, *Candidatus* Phytoplasma ziziphi, AGO, gene expression, RNA-induced silencing complex

## Abstract

The cultivation of jujube (*Ziziphus jujuba*) in China is threatened by jujube witches’ broom (JWB) disease, a devastating infectious disease associated with JWB phytoplasma (‘*Candidatus* Phytoplasma ziziphi’). In many plants, proteins in the Argonaute (AGO) family, as main components of the RNA-induced silencing complex (RISC), play important roles in RNA silencing and pathogen resistance. The jujube telomere-to-telomere genome was searched by BLAST using Arabidopsis AGOs as probes. A total of nine jujube AGO gene members were identified, with each containing the conserved N-terminal, PZA, and PIWI domains. Phylogenetic analysis revealed that the nine jujube AGOs scattered into all three *Arabidopsis* AGO clades. Expression patterns of the *ZjAGO* genes were analyzed in response to phytoplasma in transcriptome data and by qRT–PCR. The jujube–phytoplasma interaction altered the expression of jujube *AGO* genes. *ZjAGO1* and *ZjAGO8* were up-regulated in the majority of the eight sampling periods subjected to qRT–PCR analysis. In the transcriptome data, *ZjAGO1* and *ZjAGO8* were also up-regulated during the key stages 37 and 39 weeks after grafting (WAG) with phytoplasma-infected material. These two jujube *Argonaute* genes may play important roles in response to JWB phytoplasma infection.

## 1. Introduction

Gene regulation, mediated by small nucleic acid sequences, including small interfering RNA (SiRNA), microRNA (miRNA), and small interfering DNA (siDNA), is a fundamental component of many biological processes, including the response to biotic and abiotic stresses. For instance, RNA silencing, induced by small RNAs, is a major pathway by which eukaryotes resist pathogen infection. RNA silencing involves four processes, including the formation of double-strand RNA (dsRNA), the generation of short RNA, the recruitment of the RNA-induced silencing complex (RISC), and the degradation of homologous mRNA [1,2].

RNA-binding proteins in the Argonaute (AGO) family are a main component of the RISC. AGO proteins have diverse functions but share a conserved structure, generally consisting of four domains. The N-terminal domain may be related to the isolation of small RNA. The PAZ domain may recognize prominent 3′ ends of sRNA and bind to single-stranded RNA. The MID domain mainly binds to the 5′-phosphate end of sRNA, anchoring sRNA to the AGO protein. The PIWI domain is structurally similar to RNase H and can degrade target mRNA. With this conserved functional structure, the AGO proteins play important roles in the biogenesis of small regulatory RNAs, in regulating gene expression, and in defending plants against pathogens via small RNA-mediated gene silencing [3,4].

Chinese jujube (*Ziziphus jujuba*), also called Chinese date, red date, or zao, is a fruit tree in the Rhamnaceae family. Jujube fruits are prized for their delicious flavor and nutrient profile. The jujube fruits are consumed fresh or are dehydrated or processed into syrups, marmalades, sweets, flour, or cider. Jujube is one of the most commonly used fruits in Chinese traditional herbal medicine [5,6].

The most destructive disease of jujube is jujube witches’ broom (JWB), associated with the presence of JWB phytoplasma (‘*Candidatus* Phytoplasma ziziphi’). JWB causes heavy losses in fruit yield and quality [7]. High-throughput transcriptome and proteome analyses have revealed that genes involved in the jasmonic acid (JA) and salicylic acid (SA) pathways respond to JWB phytoplasma infection [8,9,10]. Members of some gene families, such as bZIP [11], SQUAMOSA-like (SPL) [12], and TCP [13] transcription factors, as well as lipoxygenases (LOXs) [14], may also be involved in the jujube–phytoplasma interaction.

In the current study, we identified the jujube Argonaute gene family members from the telomere-to-telomere (T2T) genome and analyzed the *ZjAGO* expression patterns by transcriptome data mining and qRT-PCR analysis. Transcriptome data during the phytoplasma infection progress by grafting diseased buds on healthy plants and elimination progress by tetracycline treatment on in vitro diseased plantlets were analyzed. Quantitative RT-PCR was performed on samples from different plant tissues and over the summer growing season. Key jujube *AGO* genes active during jujube–phytoplasma interaction were identified.

## 2. Materials and Methods

### 2.1. Identification of AGO Genes in Jujube Genome

Ten *Arabidopsis* AGO protein sequences were downloaded from the TAIR database (http://www.arabidopsis.org) [15] and used as queries to run Protein BLAST in the NCBI *Ziziphus jujuba* ‘Dongzao’ genome (PRJNA1014986) [16]. The BLAST results were aligned by DNAMAN v6.0.3.99 to remove duplicate and redundant sequences. An analysis of conserved domains was performed by Pfam domain prediction online tools at (https://www.ebi.ac.uk/Tools/pfa/pfamscan) on 28 April 2024 [12,14].

### 2.2. Bioinformatics Analysis of the Jujube AGOs

The physicochemical properties of the ZjAGO proteins, including the number of amino acids, molecular weight (MW), and theoretical isoelectric point (pI), were predicted using the ExPASy ProtParam tool (https://web.expasy.org/protparam/) on 28 April 2024. The subcellular localization of each protein was predicted by the CELLO v2.5 online tool (http://cello.life.nctu.edu.tw). The chromosomal locations of the *ZjAGO* genes were obtained from the NCBI genome annotation file and visualized by MapInspect v1.0 software. A phylogenic tree of jujube and Arabidopsis AGO proteins was constructed by MEGA X software and visualized by iTOL (https://itol.embl.de/) on 29 April 2024. The conserved motifs of the jujube AGO proteins were analyzed by MEME v5.5.6 (https://meme-suite.org/meme/index.html) and mapped using TBtools software v 2.110 [12,13].

### 2.3. Transcriptome Mining of ZjAGOs During Phytoplasma Infection and Elimination

Two transcriptome sequencing data sets, one derived from jujube responding to JWB phytoplasma infection by grafting diseased buds on healthy plants and one during JWB phytoplasma elimination by tetracycline treatment on in vitro diseased plantlets, were previously generated by our group [8,10]. JWB phytoplasma infection (I) was initiated by grafting diseased buds onto healthy 2-year-old jujube plants. Leaves from three infected plants grafted with JWB diseased buds and three healthy control plants grafted with health buds were collected at six periods, namely 0 weeks after grafting (WAGs), 2 WAG (phytoplasma was first detected on grafted jujube trees), 37 WAG (new leaves completely emerged next spring), 39 WAG and 48 WAG (both time periods where typical JWB symptoms can be observed), and 52 WAG (jujube trees were completely infected by phytoplasma) [8,9].

To treat JWB-infected plants for elimination (E), JWB-diseased jujube plantlets were cultured in vitro in media with 25 mg/L tetracycline. Shoots from three jars were collected after 0, 1, 3, and 6 months of tetracycline treatment (MTT). Samples of 0 MTT were the control. After 3 MTT, the diseased plantlets showed signs of recovery, yet the phytoplasma 16S rRNA was still detectable. At 6 MTT, visible disease symptoms had disappeared, and the plantlet turned to PCR negative. After 9 MTT, all signs of the phytoplasma were completely eliminated from the jujube plantlets [10,14].

The relative abundance of each transcript was normalized as the value of fragments per kilobase of exon per million mapped fragments (FPKM) of the log2 fold change between control and treatment groups. The heatmap was generated by TBtools software v2.110 [12].

### 2.4. Plant Materials and Quantitative Real-Time PCR

Eight-year-old jujube trees were maintained in the experimental orchard of Henan Agricultural University in a suburb of Zhengzhou city. All sample trees were certified as healthy (H; phytoplasma negative) or diseased (D; phytoplasma positive with typical JWB symptoms) by PCR with the phytoplasma universal primer pair U3/U5 [17]. Spatial expression of *ZjAGOs* was analyzed in early June by sampling the young leaves (newly expanding leaves, HYL/DYL), old leaves (fully expanded leaves, HOL/DOL), young shoots (bearing shoots with expanding leaves, HYS/DYS), old shoots (bearing shoots with fully expanded leaves, HOS/DOS), flowers (HF/DF), phloem (HP/DP), and xylem (HX/DX) of the woody primary shoot and the whole plant (HWP/DWP) containing all of the above samples. The bearing shoots were removed from leaves and flowers and divided into leaf, flower, and shoot samples. Young primary shoots were debarked for phloem and xylem sample collection. Phloem samples were scraped from the inner part of the bark, while xylem samples were collected by scraping the surface of debarked stems [18]. The expression dynamics of the *ZjAGOs* were analyzed in a mix of leaf samples from whole bearing shoots of healthy (H) and diseased (D) jujube trees in 8 sampling periods, twice in June, July, August, and September. Primers for the 9 *ZjAGO* genes were designed with Primer Premier 5.0, as listed in Appendix A. The jujube *actin* gene was used as a reference gene for data normalization. qRT-PCR was conducted according to Ye et al. [8]. Each experiment was repeated three times with three biological replicates.

### 2.5. Subcellular Localization of ZjAGOs

Three ZjAGO proteins, ZjAGO3, ZjAGO6, and ZjAGO9, were selected to verify the prediction results. The selected *ZjAGO* genes were recombined into the pSKA277-GFP vector with green fluorescent protein (GFP) and then transferred into the *Agrobacterium tumefaciens* strain GV3101 for infiltration into tobacco (*Nicotiana benthamiana*) leaves. Primers for gene clones and subcellular location vector constructs are listed in Appendix A. The leaves were cultured in the dark for 12 h and then under light for 12 h. The fluorescence signal was observed and photographed using a Nikon A1 laser confocal microscope (Nikon Corporation Inc., Tokyo Metropolis, Japan) [19].

## 3. Results

### 3.1. Identification and Genomic Location of AGO Genes in Jujube

Using ten Arabidopsis AGO amino acid sequences as queries, a genome-wide BLAST against the jujube telomere-to-telomere (T2T) genome database (PRJNA1014986) resulted in twenty-three AGO-like protein sequences. Duplicate and redundant sequences were removed based on sequence alignment by DNAMAN software v6.0.3.99. Domain analysis predicted by Pfam online software was used to eliminate sequences without the typical AGO conserved PAZ and Piwi domains. Finally, nine *ZjAGO* genes were obtained and named from *ZjAGO1* to *ZjAGO9* according to their position on the jujube chromosomes (Table 1, Figure 1).

The physical locations of the *ZjAGO* genes were mapped onto the jujube chromosomes using MapInspect software. The nine *ZjAGO* genes were unevenly distributed on 7 of the 12 jujube chromosomes. Two chromosomes, Chr4 and 10, each contained two *ZjAGO* genes, while the other five chromosomes, Chr1, 2, 3, 11, and 12, contained one *ZjAGO* gene each. There were no *ZjAGO* genes mapped to chromosomes 5, 6, 7, 8, or 9. The *ZjAGO* genes were scattered in the jujube genome, and no tandem genes were found (Figure 1).

### 3.2. Characteristics of Jujube AGO Proteins

The characteristics of the nine jujube AGO genes and proteins were analyzed by bioinformatic techniques (Table 1). The length of the coding sequences (CDSs) ranged from 2706 bp (*ZjAGO5*) to 3246 bp (*ZjAGO1*), and the corresponding protein sequence ranged from 901 to 1081 amino acids (aa). The minimum molecular weight was 100.6 kDa (ZjAGO5), and the maximum was 120.0 kDa (ZjAGO1). The isoelectric points ranged from 9.07 (ZjAGO1) to 9.62 (ZjAGO6). All nine jujube AGO proteins were predicted to localize to the cell nucleus (Table 1).

### 3.3. Structural Analysis of ZjAGO Proteins

The conserved domains of the jujube AGO proteins were predicted and visualized using Pfam and SMART software. The conserved Argonaute N-terminal, PAZ, and Piwi domains were found in all nine ZjAGOs. The Mid domain was found in ZjAGO2, ZjAGO3, ZjAGO7, and ZjAGO8. A Gly-rich domain was found in ZjAGO2 (Figure 2a).

MEME v 5.5.6 online software identified 10 conserved motifs in the ZjAGO proteins. All nine jujube AGOs contained all 10 conserved motifs, and their relative positions were consistent. Some ZjAGOs contained more copies of certain motifs; for instance, ZjAGO1 contained two copies of motif 5, ZjAGO2 contained two motif 6 sequences, and ZjAGO4 contained two motif 4 sequences (Figure 2b).

### 3.4. Phylogenetic Analysis of ZjAGOs and AtAGOs

A phylogenetic tree of *Arabidopsis* and jujube AGO proteins was constructed by MEGA X software. The ten Arabidopsis and nine jujube AGO proteins were divided into three groups, with the groups anchored by AGO1, AGO4, and AGO7, respectively.

AtAGO1, AtAGO5, AtAGO10, ZjAGO3, ZjAGO4, ZjAGO5, and ZjAGO6 formed the AGO1 group. The AGO4 group included AtAGO4, AtAGO6, AtAGO8, AtAGO9, ZjAGO1, and ZjAGO2. AtAGO2, AtAGO3, AtAGO7, ZjAGO7, ZjAGO8, and ZjAGO9 clustered into the AGO7 group (Figure 3).

### 3.5. Transcriptome Analysis of ZjAGO Expression in Response to Phytoplasma Infection and Elimination

To reveal the responses of the jujube AGO genes to phytoplasma infection, the FPKM values of the ZjAGO transcripts during phytoplasma infection and elimination, available in our previous transcriptome data sets, were analyzed. All nine ZjAGO genes were expressed during both the infection and elimination experiments (Figure 4). In the infection experiment, *ZjAGO1* was significantly up-regulated at 37 WAG. *ZjAGO5* was significantly up-regulated at 37 WAG and 39 WAG. *ZjAGO2*, *ZjAGO6*, and *ZjAGO9* were significantly up-regulated at 39 WAG. *ZjAGO3*, *ZjAGO4*, *ZjAGO7*, and *ZjAGO8* were significantly up-regulated at 37 WAG, 39 WAG, and 48 WAG (Figure 4a).

In the phytoplasma elimination treatment, gene expression before tetracycline treatment (at 0 MTT) was set as the control. Several *ZjAGO* genes were differentially expressed at three time points after tetracycline treatment. *ZjAGO2*, *ZjAGO3*, *ZjAGO5*, *ZjAGO6*, *ZjAGO8*, and *ZjAGO9* were down-regulated 9 months after tetracycline treatment (MTT). Other *ZjAGO* genes were up-regulated, namely *ZjAGO1* at nine MTT and *ZjAGO4* and *ZjAGO7* at six MTT (Figure 4b). The gene showing the greatest decrease in expression was *ZjAGO3* at three, six, and nine MTT, with log2 fold change values of −0.96, −1.20, and −1.50. The gene with the greatest increase in expression was *ZjAGO4* at three, six, and nine MTT, with log2 fold change values of 0.15, 0.40, and 0.19 (Figure 4b).

### 3.6. Quantitative RT-PCR Analysis of ZjAGO Genes in Diseased and Healthy Jujube Trees

To further clarify the responses of the *ZjAGO* genes to phytoplasma, the expression of the nine ZjAGOs in different organs of healthy and diseased jujube trees were analyzed throughout the growing season by quantitative RT-PCR.

Eight samples, namely young leaves (HYL/DYL), old leaves (HOL/DOL), young bearing shoots (HYS/DYS), old bearing shoots (HOS/DOS), flowers (HF/DF), phloem (HP/DP), xylem (HX/DX), and the whole plant (HWP/DWP) were collected later in June. Despite ZjAGO5 and ZjAGO8, which were down-regulated in many organs in diseased trees compared to healthy trees, all other seven ZjAGOs showed an up-regulated pattern in the diseased trees. Among them, ZjAGO6 was significantly up-regulated in all eight samples of the diseased trees. ZjAGO3 and ZjAGO9 were significantly up-regulated in six samples. ZjAGO1, ZjAGO2, and ZjAGO4 were significantly up-regulated in five samples, while ZjAGO7 was up-regulated in four samples. The expressions of these seven up-regulated genes in the phloem of new primary shoots of the diseased trees are all higher than healthy ones, in which the expression of ZjAGO2 and ZjAGO4 in diseased phloem is only slightly higher than healthy ones, while the other 5 five ZjAGOs are significantly up-regulated in diseased phloem. At the same time, the expression of these seven ZjAGOs in old leaves are all significantly higher in diseased trees than healthy ones (Figure 5).

Samples were also taken from orchard-grown trees over the growing season. Leaves from bearing shoots of healthy and diseased jujube trees were collected eight times from early June to late September. The nine *ZjAGO* genes showed different expression patterns at the eight time points. Among them, *ZjAGO1* and *ZjAGO8* were the most up-regulated genes in the diseased trees. The expression of *ZjAGO8* in seven of the eight periods, except for early July, were significantly higher in the diseased trees than in healthy trees. *ZjAGO1* was higher at six time points in the diseased samples. *ZjAGO5*, *ZjAGO7*, and *ZjAGO9* were the most down-regulated genes in the diseased trees. Of these, *ZjAGO9* was significantly lower at five of the time points in the diseased trees compared to the healthy ones. *ZjAGO5* and *ZjAGO7* were significantly lower at four time points in the diseased trees (Figure 6).

### 3.7. Subcellular Localization of Jujube AGO Proteins

To verify the subcellular location predictions, three jujube AGOs, namely ZjAGO3, ZjAGO6, and ZjAGO9, were selected for subcellular localization analysis using transiently expressed GFP fusion proteins. The expression vectors pSAK277-ZjAGO3-GFP, pSAK277-ZjAGO6-GFP, and pSAK277-ZjAGO9-GFP were constructed, transformed into *Agrobacterium*, and infiltrated into *Nicotiana benthamiana* leaves. After three days of transient expression, the GFP and mCherry fluorescence signals were observed. Each of the three AGO proteins were detected in the cell nuclei (Figure 7). These results were consistent with the predicted localizations and imply that these AGO proteins may function in the nucleus.

## 4. Discussion

### 4.1. Identification of the Jujube AGO Family Members

Jujube was among the first deciduous fruit tree species to undergo genome sequencing [20,21]. Recently, telomere-to-telomere (T2T) gapless genome assemblies have also been accomplished in a few jujube cultivars [16,22,23]. A combination of these domesticated jujube genomes with wild jujube genome data has made pan-genome analyses on jujube possible, leading to the discovery of candidate genes involved in flowering regulation and fruit ripening [23,24]. These genome assemblies, along with high-throughput genome resequencing [25,26] and transcriptome sequencing [9,10,20], provide plenty of data for genetic analysis. Many gene family members, such as the APETALA2/ethylene response factor (AP2/ERF) [27], β-Amylase (BAM) [28], cellulose synthase (Ces/Csl) [29], lipoxygenase (LOX) [14], and transcription factor families such as bHLH [30], bZIP [11], MYB [31], NAC [32], SPL [12], TCP [13], and WRKY [33] have been identified and preliminarily characterized during fruit development or in response to biotic/abiotic stressors. Some members of the bZIP, LOX, SPL, and TCP gene families were tagged as playing roles in the jujube–phytoplasma interaction [11,12,13,14].

Members of the Argonaute family of RNA-binding proteins have been identified in many plant species. For example, there are 10 AGO family members identified in the Arabidopsis genome [34], 15 in apple (*Malus* × *domestica*) [35], 13 in grapevine (*Vitis vinifera*) [36], 18 in maize (*Zea mays*) [37], 15 in poplar (*Populus trichocarpa*) [38], 19 in rice (*Oryza sativa*) [39], and 15 in walnut (*Juglans rega*) [40]. Using the latest jujube T2T gapless genome data, we identified nine jujube AGO family members. This is a smaller number of gene family members compared with other plant species. There are no gene clusters or tandem genes for AGO in the jujube genome, implying that no duplication events occurred. These results provide useful information for the further elucidation of AGO functions in jujube.

### 4.2. Phylogenetic Analysis of Arabidopsis and Jujube AGOs

In Arabidopsis, the 10 AGO proteins are divided into three phylogenetic clades, namely AGO1/AGO5/AGO10, AGO2/AGO3/AGO7, and AGO4/AGO6/AGO8/AGO9 [41]. Diverse functions of Arabidopsis AGOs have been deduced from the nature of the sRNAs they can bind. AGO1 associates with most miRNAs and a variety of siRNAs, mainly regulates plant development and stress adaptions, and contributes to pathogen-associated molecular pattern-triggered immunity (PTI) [42,43]. AGO2 plays a role in antibacterial immunity by binding to miR393b* [44]. AGO7 predominantly binds to miR390, triggers the generation of trans-acting siRNAs, and contributes to effector-triggered immunity (ETI) [45,46]. AGO4, AGO6, and AGO9 all bind to endogenous 24-nt sRNAs and function in RNA-directed DNA methylation [47].

In the present study, nine jujube AGO family members clustered into the same three AGO clades. The ZjAGO members in a clade shared similar gene structures. For example, ZjAGO5 and ZjAGO6 in the AGO4 clade lack the ArgolMid domain. ZjAGO1, ZjAGO4, and ZjAGO9 in the AGO7 clade lacked both the Argol2 and ArgolMid domains. The presence of the conserved domains and motifs indicated that the structure and function of jujube AGOs are relatively conserved. Since functions of most of the Arabidopsis AGO proteins have been revealed by previous studies, it is likely that the roles of the jujube AGOs could be predicted by the function of the corresponding Arabidopsis AGO proteins in the same clade.

### 4.3. AGOs in Jujube–Phytoplasma Interaction

Phytoplasma infection and colonization modulate host plant gene expression patterns. Through transcriptomic, proteomic, and qRT–PCR analyses, many transcription factors and functional genes have been identified as active during jujube–phytoplasma interactions. Most of these genes are involved in biotic stress responses, phytohormone biosynthesis, or metabolic pathways [7].

In the present study, the mining of jujube transcriptome data revealed that the expression of the AGO genes are also affected by phytoplasma infection or elimination. The qRT–PCR analysis of jujube AGO gene expression over the growing season revealed that ZjAGO1 and ZjAGO8 were significantly up-regulated. This is consistent with the up-regulated expression patterns at 37 and 39 WAG, which are believed to be key stages during the phytoplasma infection [8,9]. Phytoplasmas are sensitive to tetracycline. In the phytoplasma elimination experiment, at 9 months of tetracycline treatment (9MTT) when all signs of phytoplasma were completely eliminated, ZjAGO1 showed the most up-regulated expression, while ZjAGO8 showed significant down-regulation (Figure 4). Changes in the ZjAGO1 and ZjAGO8 expression after nine MTT compared with 0 MTT also imply they are involved in phytoplasma–plant interaction. These two genes may play an important role in jujube responses to JWB phytoplasma.

RNA silencing induced by microRNAs is a general strategy for plants to resist pathogens. The expression of miRNA in jujube has been proven to be influenced by the infection of phytoplasma. Twelve up-regulated and ten down-regulated miRNAs were identified in JWB-infected jujube plants [48]. As main components of the RNA-induced silencing complex (RISC), the functions of jujube Argonaute proteins during phytoplasma infection need further research. An important first step would be to test the binding of these 22 miRNAs by the differentially regulated AGO proteins in jujube.

## 5. Conclusions

We identified nine AGO gene family members in the latest jujube T2T genome data set. These *ZjAGO* genes were scattered across three phylogenetic clades, similar to their counterparts in *Arabidopsis*. The jujube–phytoplasma interaction changed the expression pattern of all of the jujube *AGO* genes. *ZjAGO1* and *ZjAGO8* may play important roles in the response to JWB phytoplasma infection.

## Figures and Tables

**Figure 1 microorganisms-13-00658-f001:**
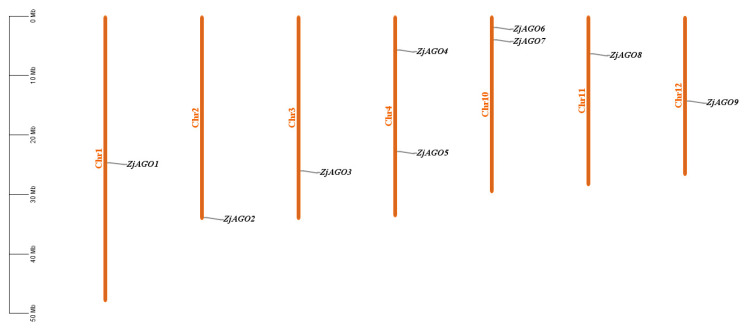
Chromosomal location of the jujube *AGO* genes. The genes were numbered based on the order of their chromosomal locations. No predicted AGO genes were mapped to chromosomes 5, 6, 7, 8, or 9.

**Figure 2 microorganisms-13-00658-f002:**
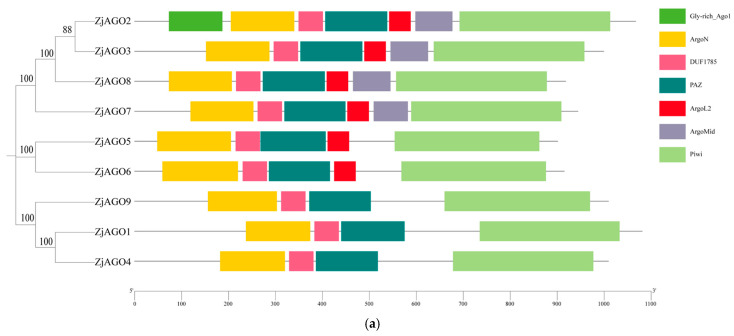
Conserved domains and motifs of the jujube AGO proteins. (**a**) Seven conserved domains (Gly-rich Ago1, ArgoN, DUF1785, PAZ, Argol2, ArgoMid, and Piwi) predicted in jujube AGOs; (**b**) protein motifs (numbered 1–10) predicted in jujube AGOs.

**Figure 3 microorganisms-13-00658-f003:**
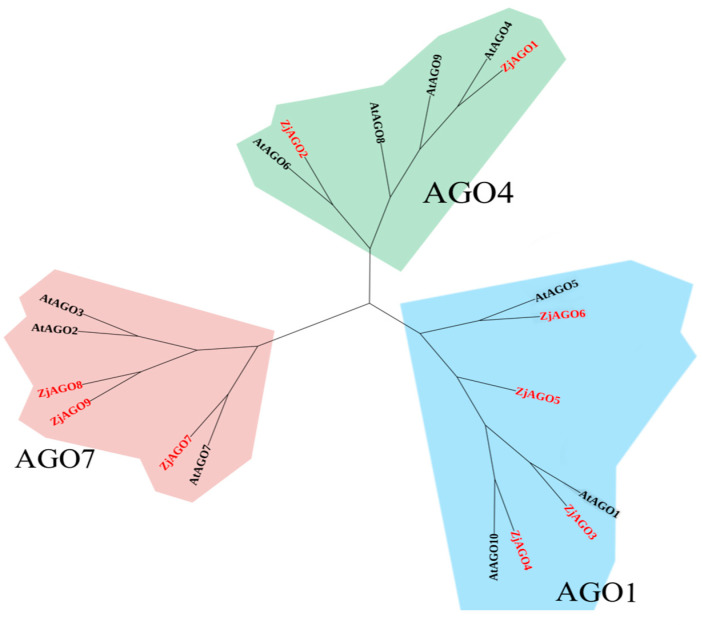
Phylogenetic tree of *Arabidopsis* and jujube AGO proteins.

**Figure 4 microorganisms-13-00658-f004:**
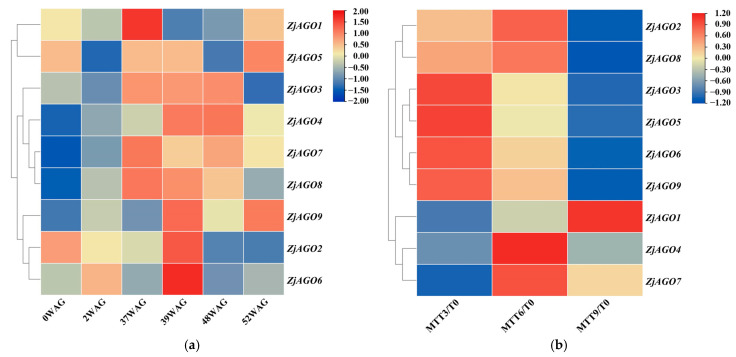
Heat map of ZjAGO expression in response to phytoplasma infection and elimination. Blue represents down-regulated genes and red represents up-regulated genes. (**a**) Transcriptome analysis of ZjAGO expression in six periods (0, 2, 37, 39, 48, and 52 WAG) after phytoplasma infection; (**b**) transcriptome analysis of ZjAGO expression of phytoplasma elimination after 3, 6, and 9 months of tetracycline treatment (MTT) compared with 0 MTT.

**Figure 5 microorganisms-13-00658-f005:**
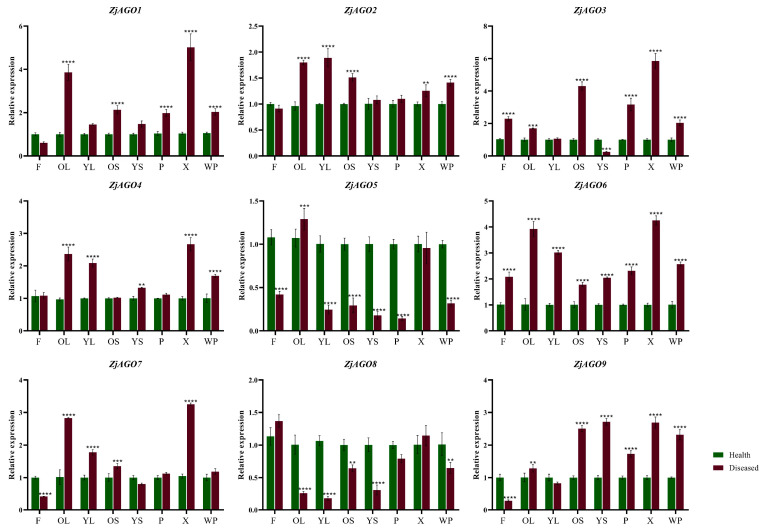
Quantitative analysis of AGO gene expression in different parts of diseased and healthy jujube trees. (F: flower; OL: old leaf; YL: young leaf; OS: old shoot; YS: young shoot; P: phloem, X: xylem; WP: whole plant). Bars represent the mean ± standard deviation of 3 plants. Asterisks indicate a significant difference between healthy and diseased samples. ** is *p* < 0.01, *** is *p* < 0.001, **** *p* ≤ 0.0001.

**Figure 6 microorganisms-13-00658-f006:**
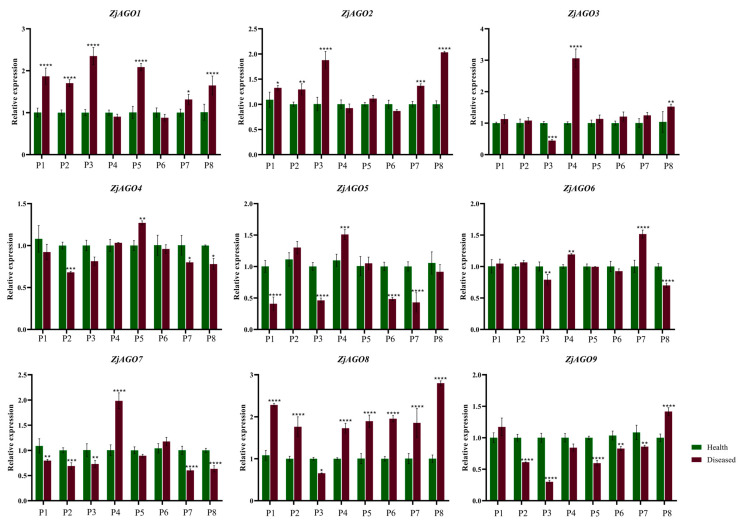
Quantitative analysis of AGO gene expression in different periods of diseased and healthy jujube leaves. (P1: early June; P2: late June; P3: early July; P4: late July; P5: early August; P6: late August; P7: early September; P8: late September). Bars represent the mean ± standard deviation of 3 plants. Asterisks indicate a significant difference between healthy and diseased samples. * is *p* < 0.05, ** is *p* < 0.01, *** is *p* < 0.001, **** *p* ≤ 0.0001.

**Figure 7 microorganisms-13-00658-f007:**
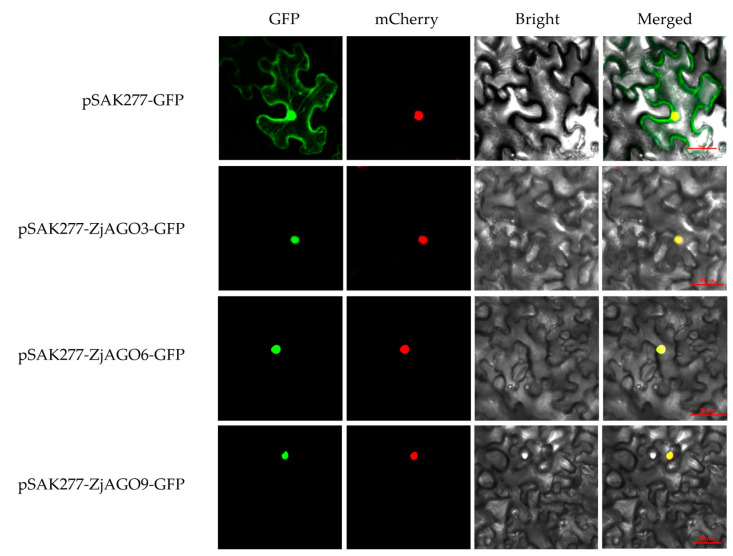
Subcellular localization of ZjAGO3, ZjAGO6, and ZjAGO9. GFP panel represents GFP fluorescence (green color); mCherry panel represents mCherry fluorescence (red color); Brightfield panel repersents bright field photograph; Merged panel represents bright field overlay of GFP and mCherry fluorescence.

**Table 1 microorganisms-13-00658-t001:** Identification of nine jujube AGO genes.

Gene Name	Gene Symbol	Length of CDS (bp)	No. of Amino Acids (aa)	Molecular Weight (Da)	Predicted Isoelectric Point (PI)	Chromosome Location	Predicted Subcellular Localization
*ZjAGO1*	LOC107404403	3246	1081	120,001.37	9.07	Chr1:24679370..24684596	Nucleus
*ZjAGO2*	LOC107419730	3204	1067	118,261.55	9.24	Chr2:33914685..33921947	Nucleus
*ZjAGO3*	LOC107423077	3000	999	112,063.38	9.39	Chr3:26040251..26050022	Nucleus
*ZjAGO4*	LOC107416305	3030	1009	113,335.91	9.24	Chr4:5706781..5712263	Nucleus
*ZjAGO5*	LOC107417145	2706	901	100,600.59	9.14	Chr4:22782106..22789558	Nucleus
*ZjAGO6*	LOC107410430	2748	915	101,978.57	9.62	Chr10:1951063..1958741	Nucleus
*ZjAGO7*	LOC107410576	2835	944	105,216.19	9.42	Chr10:4031365..4037457	Nucleus
*ZjAGO8*	LOC107431701	2757	918	104,016.88	9.34	Chr11:6332489..6340927	Nucleus
*ZjAGO9*	LOC107428912	3030	1009	114,734.97	9.31	Chr12:14329119..14333120	Nucleus

## Data Availability

The original contributions presented in this study are included in the article/Appendix A. Further inquiries can be directed to the corresponding authors.

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
