# Peer review of "Infection with Jujube Witches’ Broom Phytoplasma Alters the Expression Pattern of the Argonaute Gene Family in Ziziphus jujuba"

_microorganisms, 2025, doi:10.3390/microorganisms13030658_

Round 1
Reviewer 1 Report
Comments and Suggestions for Authors
- Line 51: fruits
- Line 64: phytoplasma elimination? Please, clarify.
- Line 85: Transcriptome
- Line 87: during phytoplasma elimination? Please, clarify.
- Lines 95-101: did you include healthy plants treated with tetracycline as control? You evaluated the differentially expressed genes in phytoplasma-infected plants versus plants treated with tetracycline for phytoplasma elimination. In my opinion, it is necessary to include healthy plants treated with tetracycline to correlate if expression differences are due effectively to phytoplasma elimination or to tetracycline effect. Moreover, two-year-old plants were used to evaluate phytoplasma infection while jujube plantlets maintained in vitro were utilized to evaluate phytoplasma elimination. Did you evaluate transcriptome differences in healthy plants grown in these different conditions? Phytoplasmas were transmitted to two-year-old plants by grafting, compared with healthy ungrafted control plants: to eliminate the possible grafting effect on plant gene expression, did you consider making grafting with healthy plant tissue on control plants?
- Line 113: How did you isolate phloem and xylem tissues?
- Lines 299-324: Discussion at point 4.3 and Conclusions, concerning differential expression of ZjAGOs, can be acceptable if authors address properly the comment at lines 95-101.
Reviewer 2 Report
Comments and Suggestions for Authors
The Authors studied genes AGO expression and function in response of ´phytoplasma infection, which is a disease of jujube tree. The manuscript sounds good, the introduction is ok as well as M&M, results and discussion. The results will high contribute in the knowledge of how AGO genes works in Arabdopisis and jujuba during infection. Nine different AGO genes were identified, and only two ZjAGO1 and ZjAGO 9 play important function in the response of JWB phytoplasma infection. These results sounds very good.
Author Response
Thank you very much for taking time to review this manuscript. We really apprciate your positive comments and feedback regarding our paper.
Round 2
Reviewer 1 Report
Comments and Suggestions for Authors
I would like to thank the authors for their reply and for the text modifications.